# Study on Dynamic Mechanical Properties of Carbon Fiber-Reinforced Polymer Laminates at Ultra-Low Temperatures

**DOI:** 10.3390/ma16072654

**Published:** 2023-03-27

**Authors:** Wenhao Zhao, Sanchun Lin, Wenfeng Wang, Yifan Yang, Xuan Yan, Heng Yang

**Affiliations:** 1State Key Laboratory of Explosion Science and Technology, Institute of Advanced Structure Technology, Beijing Institute of Technology, Beijing 100081, China; 2Beijing Institute of Aerospace Systems Engineering, Beijing 100076, China

**Keywords:** carbon fiber-reinforced polymer, ultra-low temperature, dynamic mechanical properties, finite element analysis

## Abstract

This study uses experimental methods, theoretical research, and numerical prediction to study the dynamic mechanical properties and damage evolution of CFRP laminates at ultra-low temperatures. Based on the Split Hopkinson Pressure Bar (SHPB) device, we set up an ultra-low temperature dynamic experimental platform with a synchronous observation function; the dynamic mechanical properties of laminates were tested, and the damage evolution process was observed. The experimental results are as follows: The compression strength and modulus increase linearly with the increase in strain rate and show a quadratic function trend of increasing and then decreasing with the decrease in temperature. The damage degree of the dynamic bending sample increases obviously with the impact velocity and decreases first and then increases with the decrease in temperature. Based on the low-temperature dynamic damage constitutive, failure criterion, and interlayer interface damage constitutive of the laminates, a numerical model was established to predict the dynamic mechanical properties and damage evolution process of CFRP laminates at ultra-low temperatures, and the finite element analysis (FEA) results are consistent with the experimental results. The results of this paper strongly support the application and safety evaluation of CFRP composites in extreme environments, such as deep space exploration.

## 1. Introduction

Due to their remarkable advantages, such as high specific strength, elevated impact resistance, light weight, and low-cost manufacturing, CFRP composites have been widely used in critical fields, including aerospace, the automotive industry, and rail transit in recent years [1,2,3]. With the development of deep space exploration projects such as the Lunar and Mars exploration, CFRP composites used in deep space exploration face complex work environments involving ultra-low temperatures and dynamic impact [4,5]. The mechanical properties of CFRP composites under ultra-low temperatures and dynamic impact differ from those under room temperature and static environments. There may be some problems under ultra-low temperatures, such as matrix toughness deterioration and thermal expansion mismatch at the internal interface. The high strain rate impact will also cause great changes in material properties and failure forms. Extreme environmental temperature and load conditions make the service security of CFRP composites face great challenges in deep space exploration projects. Therefore, studying the dynamic mechanical properties and damage evolution of CFRP composites at ultra-low temperatures is very important for future applications.

Under ultra-low temperatures, researchers conducted static mechanical experiments on the CFRP laminates, such as tensile, compression, and three-point bending [6,7,8,9]. They found that the interface between the fibers and matrix is strengthened at low temperatures, which improves the static mechanical properties of the laminates. As a relatively mature loading method, the SHPB was used to test the dynamic mechanical properties of CFRP composites as early as 1980 [10]. Since then, researchers [11,12,13] have studied the dynamic mechanical properties of laminates using the SHPB and found that the dynamic compression shows a noticeable strain rate effect, and the compression strength and modulus are improved, to a certain extent, at high strain rates. Gomezdelrio et al. [14] study the dynamic tensile behavior of CFRP laminates at low temperatures, and the results show that temperature and strain rate have little influence on the tensile strength along the fiber direction, but the transverse strength increases obviously at low temperatures and high strain rates. Jia et al. [9] study the dynamic bending mechanical properties of CFPR laminates at −100 °C, and the results show that the laminates can withstand a greater impact load at low temperatures. It is worth noting that some researchers [15,16] used a high-speed camera to observe the damage evolution process of samples during dynamic compression experiments of composites. In general, there are few studies on the mechanical properties of CFRP laminates considering both temperature and strain rate effects, and there is also a lack of observation methods for the full-field deformation of samples at low temperatures.

The prediction of mechanical properties and the damage evolution of composites using constitutive models and FEA is an essential direction for future research. Wysmulski et al. [17,18,19] established a numerical model to predict the stability, load-carrying capacity, and damage evolution of laminates and their structural members, and the numerical and experimental results show satisfactory agreement. Some researchers introduce strain rate and temperature effect into the classical constitutive model of laminates to describe the response of CFRP laminates and use the cohesive zone model to simulate the interlayer interaction [20,21,22]. Ren et al. [23,24] developed a multi-scale finite element model for predicting the failure of composites at low temperatures, but this microstructure-based finite element model is expensive to calculate. Currently, the proposed constitutive model does not consider the strain rate and temperature effects simultaneously, resulting in a significant error between the FEA and experimental results. Although carbon fiber is independent of strain rate and temperature [25,26], experimental results show that laminates have noticeable temperature and strain rate effects. Therefore, the numerical study of the dynamic mechanical properties and damage evolution of CFRP laminates at low temperatures requires a constitutive model that considers strain rate and temperature effects.

In this paper, the dynamic mechanical properties and the damage evolution of CFRP laminates at ultra-low temperatures are studied using experimental methods, theoretical research, and numerical prediction. Firstly, based on the SHPB device, we set up an ultra-low temperature dynamic experimental platform with a synchronous observation function; the dynamic mechanical properties of CFRP laminates were tested, and the damage evolution process was observed. Secondly, the low-temperature dynamic constitutive model of CFRP laminates was established by introducing the strain rate and temperature effect into the classical constitutive model of laminates. Then, we also considered the 3D Hashin criterion and the interlayer interface damage constitutive based on the bilinear cohesive zone model. Finally, according to the theoretical model mentioned above, the numerical prediction model was established to predict the dynamic mechanical properties and damage evolution process of CFRP laminates at ultra-low temperatures, and the FEA results are consistent with the experimental results.

## 2. Experiment

### 2.1. Experimental Platform

Based on the SHPB device, we independently set up an ultra-low temperature dynamic experimental platform with a synchronous observation function. The schematic diagram of some critical equipment of the platform is shown in Figure 1. It mainly includes three parts: the SHPB device (high-pressure air pump, Hopkinson pressure bar, and digital signal collector), the ultra-low temperature system (cryogenic box, temperature controller, and liquid nitrogen pressure tank), and the synchronous observation system (synchronous signal generator, high-speed camera, and light source). The solenoid valve connecting the temperature controller controls the amount of liquid nitrogen supplied to the cryogenic box, and the temperature in the cryogenic box can be adjusted. When the real-time temperature of the cryogenic box reaches the set value, put the samples into the cryogenic box and ensure that each sample is kept at the set temperature for more than 30 min. The SHPB device is activated for dynamic impact, and the high-speed camera is triggered to shoot the whole impact process. The sample will be sprayed with speckles on the photographed side for deformation field analysis.

### 2.2. Material Preparation and Experimental Scheme

The quasi-isotropic CFRP laminates studied in this paper are prepared from carbon fiber produced by Toray Corporation of Japan and TDE-85 epoxy resin made in China using autoclave process [7]. The specific preparation process is as follows: First, cut the prepreg and place it in the mold according to the layup direction [0/45/90/−45/0/45/90/−45/0/45/90/−45]_2s_. Then, vacuum package the mold containing prepreg. Finally, push the mold into the autoclave and cure it according to the specific curing curve (the maximum curing temperature is 180 °C, and the maximum pressure is 0.6 MPa in the autoclave). Some studies have pointed out that the laminate samples used in dynamic compression experiments are cylinders or cuboids, which has little influence on the final experimental results, but the length–diameter ratio is suitable in the range of 0.5~2.0. Considering the processing difficulty, the thickness of the original laminates, and the diameter of the Hopkinson pressure bar, the dynamic compression sample is finally processed into a cuboid of 10 mm×10 mm×8 mm, and the surface of each sample needs to be polished to ensure it is smooth and parallel. According to the requirements of the composite material three-point bending test standard GB/T1449-2005 [27] and considering the small size of the cryogenic box and the support of the Hopkinson bending bar, the dynamic bending sample is finally processed into a cuboid of 60 mm×10 mm×2 mm. The shapes of the two samples are shown in Figure 2.

CFRP composites for deep space exploration need to operate in ultra-low temperatures, such as −233 °C~123 °C on the surface of the moon and −183 °C~127 °C on the surface of Mars, and the self-designed cryogenic box system can only achieve the lowest temperature of −180 °C, so we chose three temperatures (RT,−80 °C, and−180 °C) to study the mechanical properties of CFRP laminates at low temperatures. The dynamic compression experiments of three strain rates (800/s, 1200/s, and 1600/s) and three temperatures (RT,−80 °C, and−180 °C) were carried out for each group of dynamic compression samples. The dynamic bending experiments of three impact velocities (the impact velocity was selected according to three damage degrees of the sample: slight damage, obvious damage, and complete fracture) and three temperatures (RT,−80 °C,and −180 °C) were carried out for each group of dynamic bending samples. Each group of experiments was repeated three times to prevent excessive deviations or data errors and ensure the reliability of the experimental results.

## 3. Simulation

### 3.1. Low-Temperature Dynamic Constitutive Model

There is a weak coupling between temperature effects and strain rate effects of CFRP composites [9], indicating that the effects of strain rate and temperature can be calculated separately and superimposed. Research shows that the mechanical properties of resin matrix are sensitive to temperature, which is the main reason for the strength and modulus of laminates changing with temperature [28,29]. Assuming that the single-layer plate is orthotropic, the relationship between its mechanical properties in each direction and temperature can be described by the following linear expressions:(1a)$$E_{i}^{T}=\left(k_{i}T+b_{i}\right)E_{i},\;i=1,2,3$$
(1b)$$Y_{j}^{T}=\left(k_{j}T+b_{j}\right)Y_{j},\;\;\;j=t,c$$
where k and b are the material parameters that can be confirmed by experiment, $E_{i}$ is the elasticity modulus in different directions of the single-layer plate, and $Y_{j}$ is strength perpendicular to the fiber direction. t and c stand for tensile and compression, respectively. Xie et al. [30] study the dynamic compression mechanical properties of CFRP laminates under one-dimensional stress using SHPB devices. The stress–strain curves of the material at different strain rates were obtained, and the variation rule of the material strength with the strain rate was given. As our material system is the same as in the work by Xie et al., the coefficient of strain rate effect of composite is obtained
(2)$$\eta_{D}=\left\{\begin{array}{c} 1+A{\left(\frac{\dot{\varepsilon }}{\dot{\varepsilon_{0}}}\right)}^{n},\;\;\;if\;\dot{\varepsilon }>\dot{\varepsilon_{0}}\\ \;\;\;\;\;1\;\;\;\;,\;\;\;otherwise \end{array}\right.$$
where $\dot{\varepsilon }$ is the current strain rate of the sample, $\dot{\varepsilon_{0}}$ is the reference strain rate, and A and n are the parameters fitted by the experimental results. The modulus and strength of the single-layer plate considering the strain rate effect are as follows:(3a)$$E_{i}^{D}=\eta_{D}^{E_{i}}E_{i},\;i=1,2,3$$
(3b)$$Y_{j}^{D}=\eta_{D}^{Y_{j}}Y_{j},\;\;\;j=t,c$$
Finally, the expressions of modulus and strength of the single-layer plate considering temperature effect and strain rate effect are
(4a)$$E_{i}^{DT}=\eta_{D}^{E_{i}}\left(k_{i}T+b_{i}\right)E_{i},\;i=1,2,3$$
(4b)$$Y_{j}^{DT}=\eta_{D}^{Y_{j}}\left(k_{j}T+b_{j}\right)Y_{j},\;\;\;j=t,c$$

The corresponding stiffness matrix $C^{DT}$ can be calculated by following Equation (5):(5)$$C^{DT}=\frac{1}{a}\left[\begin{array}{cccccc} E_{1}^{DT}\left(1-\upsilon_{23}\upsilon_{32}\right) & E_{2}^{DT}\left(\upsilon_{12}-\upsilon_{23}\upsilon_{32}\right) & E_{3}^{DT}\left(\upsilon_{13}-\upsilon_{23}\upsilon_{32}\right) & & & \\ E_{1}^{DT}\left(\upsilon_{21}-\upsilon_{31}\upsilon_{23}\right) & E_{2}^{DT}\left(1-\upsilon_{13}\upsilon_{31}\right) & E_{3}^{DT}\left(\upsilon_{23}-\upsilon_{21}\upsilon_{13}\right) & & & \\ E_{1}^{DT}\left(\upsilon_{31}-\upsilon_{21}\upsilon_{32}\right) & E_{2}^{DT}\left(\upsilon_{32}-\upsilon_{12}\upsilon_{31}\right) & E_{3}^{DT}\left(1-\upsilon_{12}\upsilon_{21}\right) & & & \\ & & & aG_{12} & & \\ & & & & aG_{23} & \\ & & & & & aG_{13} \end{array}\right]$$
where $a=1-\upsilon_{12}\upsilon_{21}-\upsilon_{23}\upsilon_{32}-\upsilon_{13}\upsilon_{31}-2\upsilon_{12}\upsilon_{23}\upsilon_{31}$.

When the failure condition is reached, damage will occur, and the mechanical properties of the single-layer plate will decrease with the accumulation of damage. This study used different damage variables to control the stiffness degradation caused by different failure modes. Three damage variables $d_{f}$, $d_{m}$, and $d_{s}$ were introduced, where $d_{f}$ represents the damage along the fiber direction, $d_{m}$ is damage perpendicular to the fiber direction, and $d_{s}$ denotes the damage of shear. The damaged stiffness matrix can be calculated according to these damage variables:(6)$$C_{d}^{DT}=\frac{1}{a}\left[\begin{array}{cccccc} \left(1-d_{f}\right)C_{11}^{DT} & \left(1-d_{s}\right)C_{12}^{DT} & \left(1-d_{f}\right)C_{13}^{DT} & & & \\ & \left(1-d_{m}\right)C_{22}^{DT} & \left(1-d_{s}\right)C_{23}^{DT} & & & \\ & & C_{33}^{DT} & & & \\ & & & \left(1-d_{s}\right)C_{44}^{DT} & & \\ & & & & C_{55}^{DT} & \\ & & & & & C_{66}^{DT} \end{array}\right]$$

### 3.2. Failure Criterion and Damage Evolution

3D Hashin criterion can consider various failure modes of different components of composites [31,32,33]. For carbon fiber tension failure $\;\left(\sigma_{11}\geq 0\right)$:(7)$$f_{ft}={\left(\frac{\sigma_{11}}{X_{t}^{DT}}\right)}^{2}+{\left(\frac{\tau_{12}}{S_{12}}\right)}^{2}+{\left(\frac{\tau_{31}}{S_{31}}\right)}^{2}\geq 1$$

For carbon fiber compression failure $\;(\sigma_{11}<0)$:(8)$$f_{fc}={\left(\frac{\sigma_{11}}{X_{c}^{DT}}\right)}^{2}\geq 1$$

For resin matrix tension failure $\;\left(\sigma_{22}+\sigma_{33}\geq 0\right)$:(9)$$f_{mt}={\left(\frac{\sigma_{22}+\sigma_{33}}{Y_{t}^{DT}}\right)}^{2}+\frac{\tau_{23}^{2}-\sigma_{22}\sigma_{33}}{S_{23}^{2}}+{\left(\frac{\tau_{12}}{S_{12}}\right)}^{2}+{\left(\frac{\tau_{31}}{S_{31}}\right)}^{2}\geq 1$$

For resin matrix compression failure $\;(\sigma_{22}+\sigma_{33}<0)$:(10)$$\begin{array}{c} f_{mc}=\left(\frac{\sigma_{22}+\sigma_{33}}{Y_{c}^{DT}}\right)\left[{\left(\frac{Y_{c}^{DT}}{2S_{23}}\right)}^{2}-1\right]+{\left(\frac{\sigma_{22}+\sigma_{33}}{2S_{23}}\right)}^{2}\\ +\frac{\tau_{23}^{2}-\sigma_{22}\sigma_{33}}{S_{23}^{2}}+{\left(\frac{\tau_{12}}{S_{12}}\right)}^{2}+{\left(\frac{\tau_{31}}{S_{31}}\right)}^{2}\geq 1 \end{array}$$
where $X_{t}^{DT}$ and $X_{c}^{DT}$, respectively, are tensile and compressive strengths along the fiber direction at different temperatures and strain rates, $Y_{t}^{DT}$ and $Y_{c}^{DT}$ are transverse tensile and compressive strengths at different temperatures and strain rates, and $S_{12},S_{23},\;\mathrm{and}\;S_{31}$ are shear strengths in each direction. The material parameters take into account the influence of temperature and strain rate.

The damage evolution laws degrading the stiffness matrix will be calculated when the stress fields satisfy the 3D Hashin criterion. The damage variables can be expressed as follows:(11)$$\left\{\begin{array}{c} d_{f}=1-\left(1-d_{ft}\right)\left(1-d_{fc}\right)\\ d_{m}=1-\left(1-d_{mt}\right)\left(1-d_{mc}\right)\\ d_{s}=1-\left(1-d_{f}\right)\left(1-d_{m}\right) \end{array}\right.$$
where $d_{ft}$ and $d_{fc}$ are the damage variables of fiber tensile and compression, respectively, and $d_{mt}$ and $d_{mc}$ are the damage variables of matrix tensile and compression, respectively.
(12)$$\left\{\begin{array}{c} d_{fi}=1-\frac{1}{f_{fi}}\exp\left(1-f_{fi}\right)\left(\frac{C_{11}\varepsilon_{11}^{i}L_{c}}{W_{fi}}\right),\;\;\;i=t,c\\ d_{mi}=1-\frac{1}{f_{mi}}\exp\left(1-f_{mi}\right)\left(\frac{C_{22}\varepsilon_{22}^{i}L_{c}}{W_{mi}}\right),\;\;\;i=t,c \end{array}\right.$$
where $L_{c}$ is the characteristic length of the mesh to prevent mesh sensitivity problems, and $W_{fi}\;\mathrm{and}\;W_{mi}$ are the fracture toughness values of the fiber and matrix, respectively.

### 3.3. Bilinear Cohesive Zone Model for Interface

The interlayer interface of CFRP laminates is characterized by the bilinear cohesive zone model. The damaged constitutive relation is defined as the expressions below:(13)$$t_{i}=\left(1-d\right)K_{i}\delta_{i}\;,\;\;\;i=n,s,t$$
where d is the damage variable, $t_{n}$ is the normal traction, $t_{s}\;\mathrm{and}\;t_{t}$ are the shear tractions, $\delta_{n}$ is the normal separation, and $\delta_{s}\;\mathrm{and}\;\delta_{t}$ are the shear separations. $K_{n}$ is the interfacial normal stiffness, and $K_{s}\;\mathrm{and}\;K_{t}$ are the interfacial shear stiffness. It should be noted that normal compression does not lead to interface damage. Here, the quadratic nominal stress criterion is applied to characterize the damage initiation as follows [31]:(14)$${\left(\frac{\langle \sigma_{n}\rangle }{\sigma_{n}^{0}}\right)}^{2}+{\left(\frac{\tau_{s}}{\tau_{s}^{0}}\right)}^{2}+{\left(\frac{\tau_{t}}{\tau_{t}^{0}}\right)}^{2}=1$$
where $\sigma_{n}^{0},\;\tau_{s}^{0},\;\mathrm{and}\;\tau_{t}^{0}$ are normal, transverse shear, and longitudinal shear stresses, with $\langle \sigma_{n}\rangle =\max\;\left(0,\sigma_{n}\right)$. Once the damage of the interface is activated, the stiffness will gradually reduce to zero linearly, and the damage variable d increases linearly from zero to one. Here, Benzeggagh–Kenane (BK) damage evolution based on fracture energy is employed [34]:(15)$$G^{C}=G_{n}^{C}+\left(G_{s}^{C}-G_{n}^{C}\right){\left(\frac{2G_{s}}{G_{n}+2G_{s}}\right)}^{\eta_{BK}}$$
where $G_{n}$ and $G_{s}$ are the monitored fracture energies in the normal and shear directions, respectively, $G_{n}^{C}$ and $G_{s}^{C}$ are the corresponding critical fracture energies, and $\eta_{BK}$ is the Benzeggagh–Kenane parameter.

### 3.4. Finite Element Modeling

FEA is widely used to evaluate the mechanical properties and failure behavior of fiber-reinforced composites because it can provide full-field stress and strain distribution at a low cost. According to the actual size of the dynamic compression and bending samples, the commercial analysis software ABAQUS 2017 was used to establish the finite element model of laminates. As shown in Figure 3, the dynamic compression and bending samples were 48  and 12 layers, respectively, and the thickness of the single-layer plate was 0.167 mm. Each single-layer plate was assigned corresponding material properties and orientation. The interlayer interface property can be assigned using a bilinear cohesive zone model, and the quadratic nominal stress criterion was selected as the interface damage criterion. Explicit dynamic analysis was used to conduct the simulation of the dynamic compression and bending of the samples. It is worth noting that the Hopkinson pressure bar device was modeled in a 1:1 ratio to restore the actual dynamic loading environment and the boundary conditions. The material property of the Hopkinson pressure bar device (including bullet, incident bar, transmission bar, etc.) was $\rho =7.9\;g/cm^{3},\;E=210\;GPa,\;\upsilon =0.3,\;\mathrm{and}\;c_{0}=5155\;m/s$. In the explicit dynamic analysis module, the initial velocity of the bullet was given according to the real velocity measured (10 m/s—800/s, 15 m/s—1200/s, and 20 m/s—1600/s for dynamic compression; 7 m/s, 8.5 m/s, and 10 m/s for dynamic bending), and the simulation time corresponded exactly to the real experimental time. The contact between the sample surface and the end face of the pressure bar adopted surface-to-surface contact. We considered that the temperature does not change throughout the impact process and simply set the initial temperature field in the “Predefined Field” tool of ABAQUS. These models were meshed using linear reduced integration solid elements (C3D8R) of reduced integration technique. The mesh independence verification was carried out, and it was found that the influence of the current mesh density on the calculated results can be ignored.

According to the static mechanical experimental results of the single-layer plate at different temperatures [7], the temperature-related material parameters in the above equations can be obtained: $k_{1}=4.52\times 10^{-4},\;\;k_{2}=k_{3}=-2.96\times 10^{-3},\;\;k_{t}=k_{c}=-8.98\times 10^{-4},b_{1}=0.865,b_{2}=b_{3}=1.86,\;\mathrm{and}\;b_{t}=b_{c}=0.8$; the static material properties of single-layer plate at room temperature and the tensile properties at different temperatures are shown in Table 1 and Table 2. When the reference strain rate was defined as 0.001/s, the strength of the single-layer plate in different directions and the corresponding material parameters are shown in Table 3 [30]. The cohesive zone model property of each composite was different. The parameters of the bilinear cohesive zone model at room temperature were determined by Ref. [35] and our experimental test data, as shown in Table 4.

Based on the proposed dynamic constitutive model introducing temperature and strain rate effects, user subroutine VUMAT was developed and implemented in FEA models containing the failure criterion, damage evolution, and interlayer interface damage constitutive. The classical step-by-step iterative method in strain incremental forms was used. After each incremental step, the stress states of the elements were updated and examined under the failure criterion. When the failure criterion was met, the damage variable was called, and the element stiffness degenerated to describe the occurrence of damage. The calculation flowchart of ABAQUS calling user subroutine VUMAT is shown in Figure 4.

## 4. Results and Discussion

### 4.1. Dynamic Compression Experimental Results

Figure 5a shows the damage evolution process captured by the high-speed camera and corresponding DIC analysis results of the dynamic compression sample of CFRP laminates during compression at a 1200/s strain rate. The whole damage evolution process of the sample from the beginning of the compression load to delamination can be observed in Figure 5a, including the strain distribution, cracking time, and delamination position of the sample surface. At 50 μs, a large strain appears in the lower left corner, indicating that the crack may be initiated in one layer; at 60 μs, the crack growth between the layers accelerates; at 70 μs, the layers on both sides of the crack are entirely separated. Similar damage processes also occur in other samples at different temperatures and strain rates, but the initiation time, growth speed, and expansion degree of the cracks are different, and the final damage morphology of the other samples is shown in Figure 5b.

According to the original data of the dynamic compression experiment, the equations provided by Ref. [36] can calculate the strain, strain rate, and stress of the dynamic compression sample. Figure 6 shows the stress–strain curves of the dynamic compression sample at different strain rates and temperatures. The stress increases approximately linearly with the increase in strain at first. When the critical strain is exceeded, the samples lose their load-carrying capacity due to damage, and the stress plummets after reaching the peak stress. In addition, we can only find that strain rates and temperatures significantly affect compression strength and modulus in Figure 6, and it is difficult to see the variation trend of vital mechanical properties (compressive strength and modulus) with temperature and strain rate. Therefore, we calculated two mechanical properties (compressive strength and modulus) according to the stress–strain curve and plotted their variation trend with temperature and strain rate in Figure 7.

As shown in Figure 7a,b, the compression strength and modulus increase linearly with the increase in strain rate, indicating that CFRP laminates exhibit a noticeable strain rate effect during dynamic compression. The mechanism of the strain rate effect can be found in Figure 5b; the failure degree of the dynamic compression samples increases with the increase in strain rate at room temperature, indicating that more impact energy can be absorbed, thus improving the compressive strength of the samples. As shown in Figure 7c,d, the compression strength and modulus show a quadratic function variation trend of first increasing and then decreasing with the increase in temperature. In order to further analyze the temperature effect and damage mechanism, the morphology at the damage location of the samples after dynamic compression was characterized using scanning electron microscopy (SEM). The characterization results of Figure 8 are as follows: At room temperature, the surface of the damage location is smooth, and only a small amount of matrix can be seen, indicating that the interlayer bonding is weak and delamination easily occurs. At −80 °C, a single fiber and the matrix can be seen; this is because the contraction of the matrix at a low temperature enhances the interface bonding between the matrix and the fiber, which not only causes the sample to delaminate but also causes some fibers to suffer compression damage, and the interlayer properties are enhanced. At −180 °C, there is more matrix adhesion to the fiber, and the matrix suffers compression damage together with the fiber, indicating that the interface bonding between the matrix and the fiber is enhanced more obviously. However, the decrease in temperature makes the toughness of the matrix deteriorate, and its influence gradually begins to be greater than the increase in the interface bonding strength caused by the decrease in temperature, resulting in a decrease in compression strength at a lower temperature.

### 4.2. Dynamic Bending Experimental Results

Figure 9 shows the damage evolution process captured by the high-speed camera of the dynamic bending sample of CFRP laminates at different temperatures and impact velocities, as well as the DIC analysis results at 1 ms, and it can be seen that the maximum bending strain of the dynamic bending sample appears at the impact position of the incident bar. The damage degree of the dynamic bending sample increases obviously with the increase in impact velocity, but the temperature has little influence on the damage degree of the sample, which roughly shows that the damage degree decreases first and then increases with the decrease in temperature. According to the morphology analysis of the samples after impact, the damage degree of the dynamic bending sample varies from complete fracture to incomplete fracture to complete fracture with the decrease in temperature under the 8.5 m/s impact velocity, and then the relationship between the damage degree of the dynamic bending sample and temperature can be proved. In Figure 10, the optical microscope image of the fracture position further explains the influence mechanism of the temperature effect. Delamination is obvious at room temperature, indicating that the interlayer interface strength is lower, and it is easier to completely fracture. However, the interfacial debonding between the fiber and the matrix decreases with the temperature decrease, indicating that the interlayer strength increases. It is found that the fracture is shorter at −180 °C, and the main reason is that the matrix is more prone to brittle fracture with the temperature decrease, so the damage degree increases relative to that at −80 °C.

In the dynamic bending experiment, only the samples with an impact velocity of 10 m/s are completely fractured, so we calculated the load–displacement curve applied to this group of samples using the equations mentioned in Ref. [37], as shown in Figure 11. The load increases with the increase in displacement at first, and the load drops sharply after reaching the peak load, indicating that the bending sample is damaged or fractured at this time. When the impact displacement is identical, the load applied on the dynamic bending sample first increases and then decreases with the decrease in temperature, which is caused by the combined action of the interface strengthening and the matrix toughness deterioration caused by the decrease in temperature.

### 4.3. Numerical Prediction Results

#### 4.3.1. Numerical Prediction Results of Dynamic Compression

Figure 12 shows the FEA results of the dynamic compression samples of CFRP laminates under 1600/s strain rate compression. As shown in Figure 12a, when the impact process reaches 0.21 ms, a large strain appears in the lower left corner, indicating that the crack may be initiated in one layer; the layers on both sides of the crack are entirely separated at 0.25 ms; further impact produces a greater crack and new delamination; and buckling of the laminates is observed at 0.40 ms. Similarly, the stress nephogram of Figure 12b shows the above damage evolution law. By comparing the experimental results of the 1200/s strain rate in Figure 5a, it can be seen that the FEA results of the damage evolution process of the dynamic compression sample are consistent with the experimental results.

Figure 6a presents the FEA and experimental results of the stress–strain curves of the dynamic compression samples at room temperature and different strain rates. It can be seen that the stress–strain curves with 800/s and 1600/s strain rates are in good agreement, and the errors of compression strength are within 5%. The error of the 1200/s strain rate is large, possibly due to the deviation between the material parameters and some experimental results. Figure 6f presents the FEA and experimental results of the stress–strain curves of the dynamic compression samples at different temperatures and a 1600/s strain rate. It can be seen that the trend of the curve obtained by FEA is consistent with the experiment, and the error for compression strength is within 10%.

#### 4.3.2. Numerical Prediction Results of Dynamic Bending

Figure 13 shows the stress nephogram of the FEA results of the dynamic bending sample applied at an impact velocity of 8.5m/s. When the impact load is applied to the sample, the single-layer plate on the right side of the bending will be subjected to greater tensile stress. Once the damaged condition is reached, the right single-layer plate is damaged, then is gradually damaged by layer until complete fractures occur. The damage process of FEA is consistent with the experimental results of Figure 9.

.

Figure 11 presents the FEA and experimental results of the load–displacement curves of the dynamic bending samples at different temperatures and a 10 m/s impact velocity. It can be seen that the trend of the load–displacement curve changing with temperature is consistent with the experimental results, and the errors for peak load are within 10% at all temperatures.

## 5. Conclusions

In this study, the dynamic mechanical properties of CFRP laminates at ultra-low temperatures are systematically studied through experimental testing, a theoretical model, and simulation analysis, and the damage evolution process is discussed. There are the following important conclusions:

We independently designed the observable cryogenic box and improved the Hopkinson bending bar. Based on the SHPB device, we set up an ultra-low temperature dynamic experimental platform with a synchronous observation function; the dynamic mechanical properties of CFRP laminates at ultra-low temperatures were tested, and the damage evolution process was observed simultaneously. The experimental results are as follows: CFRP laminates exhibit a noticeable strain rate effect during dynamic compression; the compression strength and modulus increase linearly with the increase in strain rate and show a quadratic function variation trend of first increasing and then decreasing with the increase in temperature. The damage degree of the dynamic bending sample increases obviously with the increase in impact velocity and decreases first and then increases with the decrease in temperature, and the damage degree is minimum at −80 ℃.Based on the ultra-low temperature dynamic constitutive, failure criterion, and interlayer interface damage constitutive of CFRP laminates, a numerical prediction model was established to predict the mechanical properties and damage evolution process of the dynamic compression and bending of CFRP laminates at ultra-low temperatures. The predicted results of the relationship between the dynamic mechanical properties and strain rate and temperature agree with the experimental results. The FEA results of the damage evolution process of CFRP laminates are basically consistent with the experimental observations.

## Figures and Tables

**Figure 1 materials-16-02654-f001:**
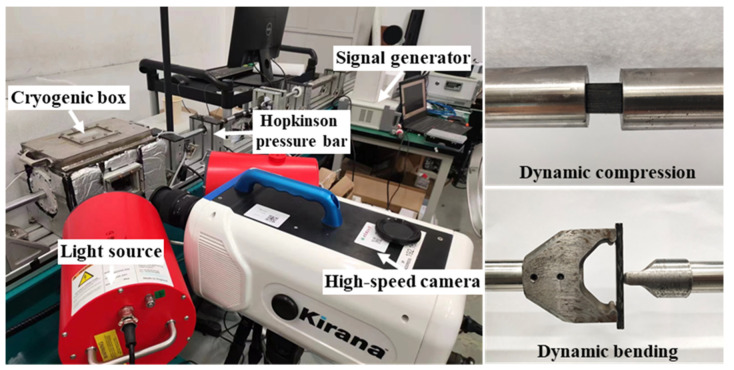
Ultra-low temperature dynamic experimental platform with synchronous observation function.

**Figure 2 materials-16-02654-f002:**
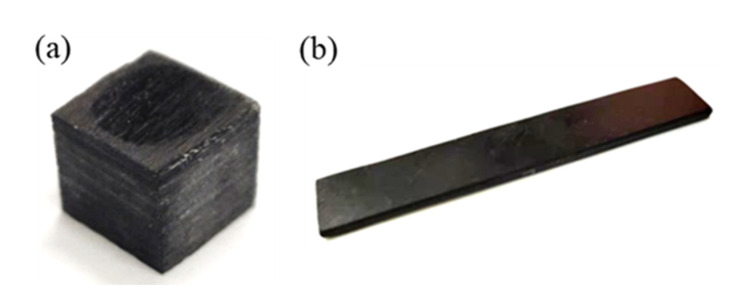
(**a**) Dynamic compression sample and (**b**) dynamic bending sample of CFRP laminates.

**Figure 3 materials-16-02654-f003:**
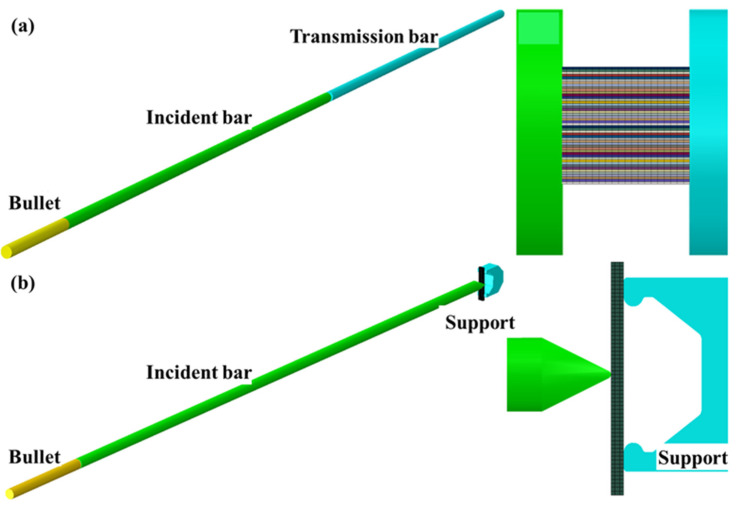
Finite element model. (**a**) Hopkinson pressure bar and dynamic compression sample. (**b**) Hopkinson bending bar and dynamic bending sample.

**Figure 4 materials-16-02654-f004:**
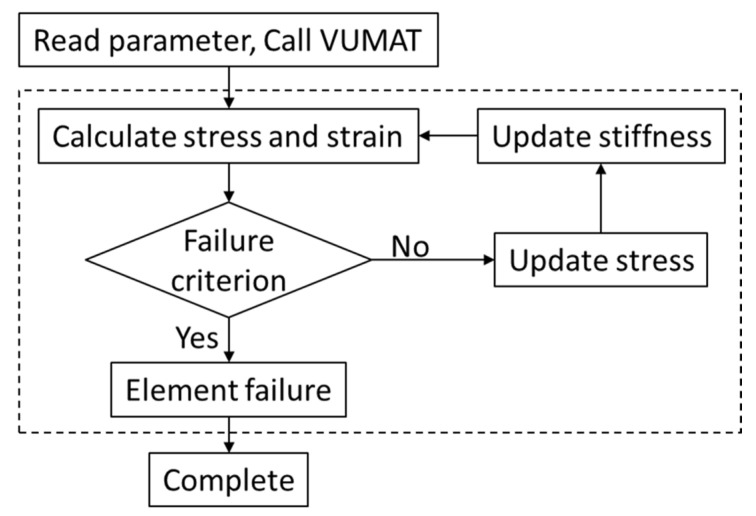
Calculation flowchart of ABAQUS calling user subroutine VUMAT.

**Figure 5 materials-16-02654-f005:**
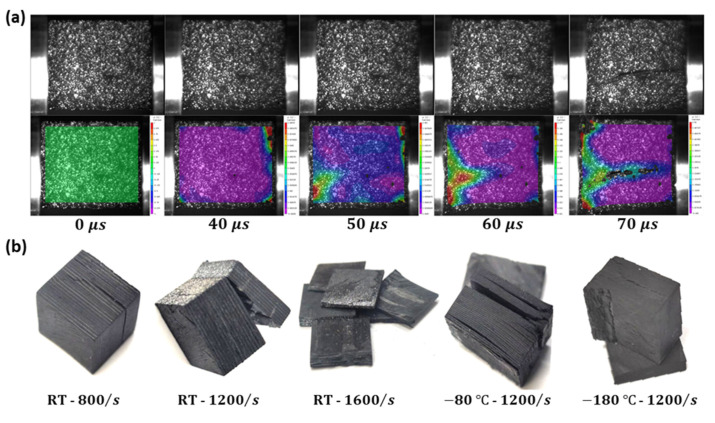
The dynamic compression samples of CFRP laminates: (**a**) the damage evolution process captured by high-speed camera and corresponding DIC analysis results during compression at a 1200/s strain rate; (**b**) the final damage morphology at different temperatures and strain rates.

**Figure 6 materials-16-02654-f006:**
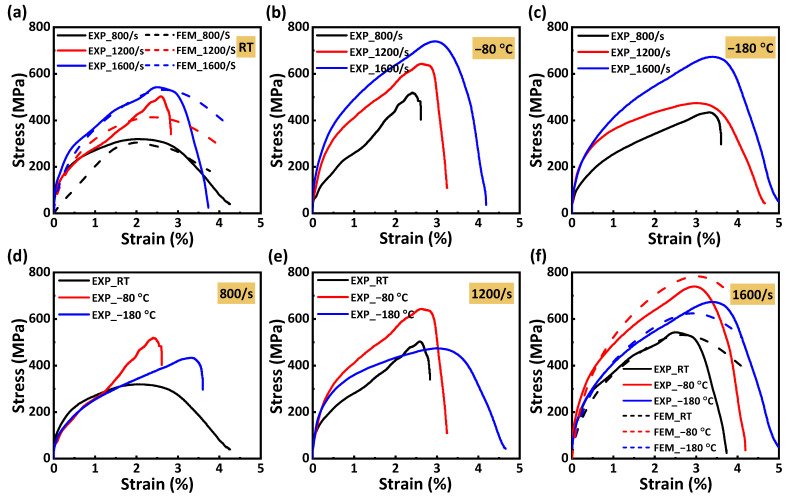
Stress–strain curves of the dynamic compression samples at different strain rates and temperatures: (**a**) room temperature; (**b**) −80 °C; (**c**) −180 °C; (**d**) 800/s strain rate; (**e**) 1200/s strain rate; (**f**) 1600/s strain rate.

**Figure 7 materials-16-02654-f007:**
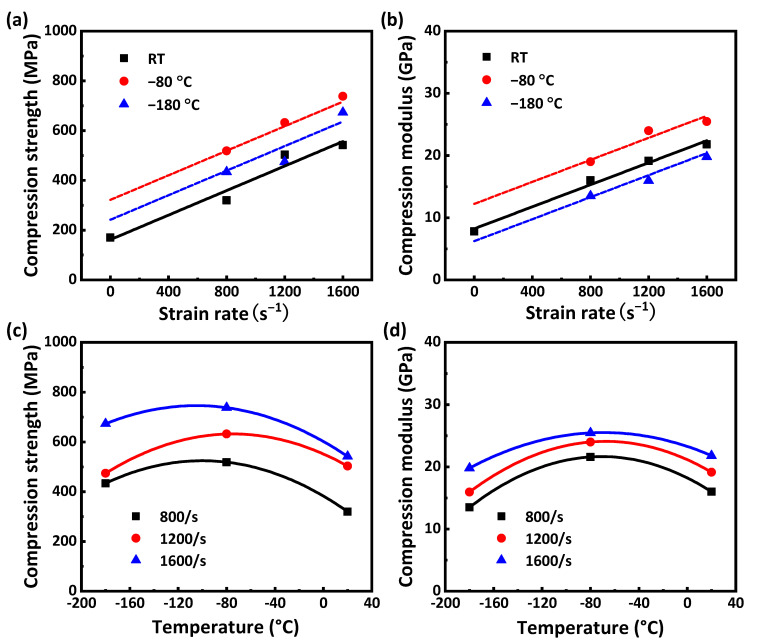
The relationship between mechanical properties of the dynamic compression sample and strain rate and temperature: (**a**) compression strength–strain rate, (**b**) compression modulus–strain rate, (**c**) compression strength–temperature, (**d**) compression modulus–temperature.

**Figure 8 materials-16-02654-f008:**
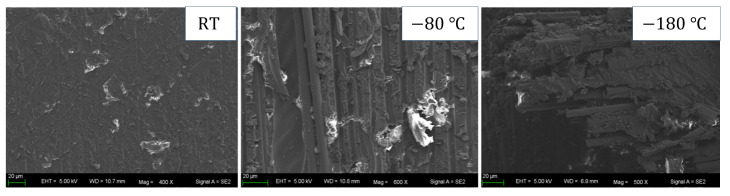
SEM image of the damage location after dynamic compression at different temperatures.

**Figure 9 materials-16-02654-f009:**
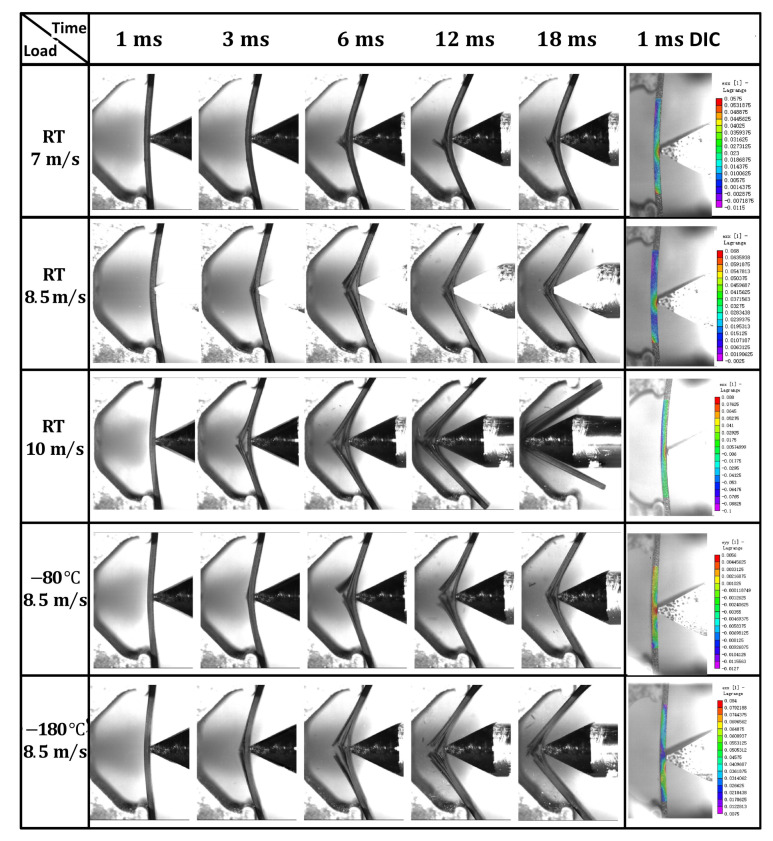
The damage evolution process captured by high-speed camera of the dynamic bending samples of CFRP laminates at different temperatures and impact velocities.

**Figure 10 materials-16-02654-f010:**
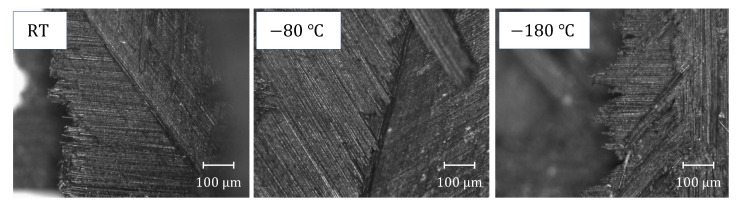
Optical microscope image of the fracture position after dynamic impact at different temperatures.

**Figure 11 materials-16-02654-f011:**
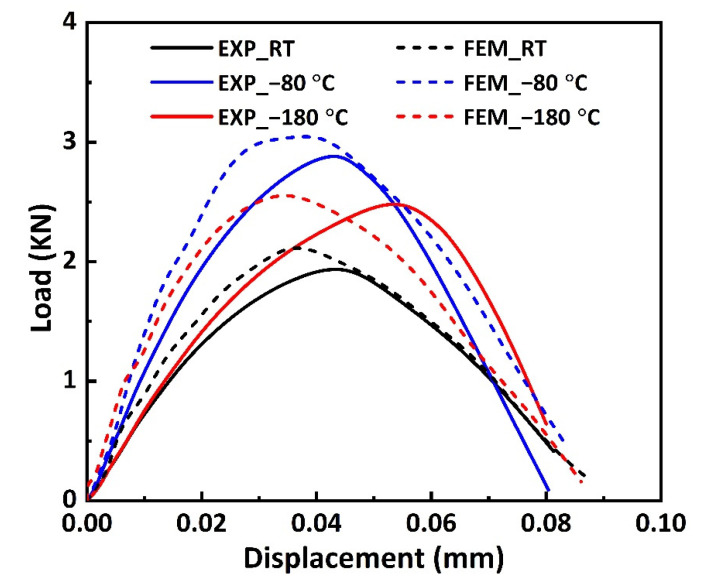
Load–displacement curves of the dynamic bending samples at different temperatures and 10 m/s impact velocity.

**Figure 12 materials-16-02654-f012:**
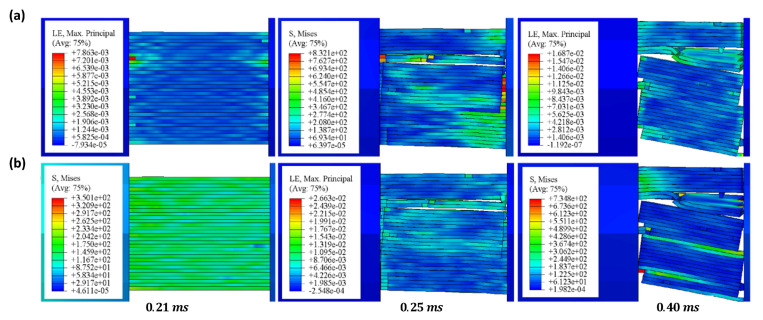
The FEA results of dynamic compression samples of CFRP laminates under 1600⁄s strain rate compression: (**a**) strain nephogram, (**b**) stress nephogram.

**Figure 13 materials-16-02654-f013:**
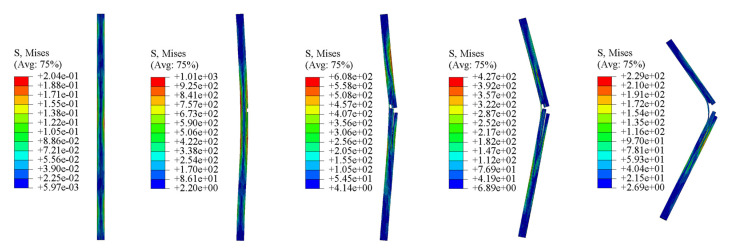
The stress nephogram of the FEA results of the dynamic bending sample applied at the impact velocity of 8.5 m/s.

**Table 1 materials-16-02654-t001:** Static material properties of single-layer plate at room temperature [7].

Modulus	$E_{1}$	$E_{2}\;=\;E_{3}$	$G_{12}\;=\;G_{13}$	$G_{23}$	$v_{12}\;=\;v_{13}$	$v_{23}$
	138 GPa	7 GPa	4.8 GPa	3.8 GPa	0.3	0.35
**Strength**	$X_{t}$	$X_{c}$	$Y_{t}$	$Y_{c}$	$S_{12}=S_{23}=S_{13}$	
	2500 MPa	800 MPa	80 MPa	150 MPa	110 MPa	
**Fracture toughness**	$W_{ft}$	$W_{fc}$		$W_{mt}$	$W_{mc}$	
	12.5 N/mm	12.5 N/mm		0.1 N/mm	0.1 N/mm	

**Table 2 materials-16-02654-t002:** Tensile properties of single-layer plate at different temperatures [7].

Samples	Temperature	Modulus $\left(GPa\right)$	Strength $\left(MPa\right)$
0°	RT	150	2181
	173 K	138	1867
	77 K	139	1828
90°	RT	8.8	56.6
	173 K	11.3	55.6
	77 K	15.4	53.2

**Table 3 materials-16-02654-t003:** The strength of the single-layer plate in different directions and the corresponding material parameters at 0.001/s reference strain rate.

	$\sigma_{0}$	A	n
Fiber direction	800 Mpa	2.04	0.41
Transverse direction	150 Mpa	4.65	0.22
Thickness direction	150 Mpa	0.11	0.50

**Table 4 materials-16-02654-t004:** The parameters of the bilinear cohesive region model at room temperature.

$K_{n}\;=\;K_{s}\;=\;K_{t}$	$\sigma_{n}^{0}$	$\tau_{s}^{0}=\tau_{t}^{0}$	$G_{n}^{C}$	$G_{s}^{C}$	$\eta_{BK}$
$1\times 10^{6}\;N/{\mathrm{mm}}^{2}$	45 MPa	60 MPa	1 N/mm	1.5 N/mm	1

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
