# Peer review of "Study on Dynamic Mechanical Properties of Carbon Fiber-Reinforced Polymer Laminates at Ultra-Low Temperatures"

_materials, 2023, doi:10.3390/ma16072654_

Round 1

Reviewer 1 Report

The review entitled "Study on dynamic mechanical properties of carbon fiber reinforced polymer laminates at ultra-low temperature” used experimental methods, theoretical research, and numerical prediction to study the dynamic mechanical properties and the damage evolution of CFRP laminates at an ultra-low temperature. An ultra-low temperature dynamic experiment platform with a synchronous observation function was established using the Split Hopkinson Pressure Bar device.

The manuscript is well-written and provides valuable experimental and numerical results. Moreover, the novelty was highlighted in the manuscript. Therefore, it should be accepted for publication after considering these two comments:

More information about the static material properties of the used CFRP laminates and epoxy resin should be provided or listed in a table.

Lines 275-277: More discussion on Figure 6 should be added. What were these significant effects?

Reviewer 2 Report

Though the work is unique, the quantum or scope of the work sounds very limited. Moreover, a number of important findings are presented without proper justifications. Hence, the following comments are to be considered  while preparing the revised manuscript:

1. Abstract is to be made concise by reducing the redundant information and present it within 150-200 word limit.

2. Please clarify why the authors have evaluated only the phenomenon for CFRP laminates? Why not the other FRP's? Glass and Basalt FRP sections are also commonly used now a days. Please clarify.

3. Regarding the ultra-low-temperature tests, please clarify the rationale behind the selection of temperature range? Why a value of -180 degree is selected?

4. Also clarify, how the low-temperature range was achieved? Mechanism of temperature variation over time is to be better explained.

5. Section 3 till 3.3. looks irrelevant to the contribution of the proposed work. Hence, they are to be summarised in the revised submission.

6. It is to be clarified, how the low-temperature profile is inputted into the ABAQUS software for FE analyses. A lot of important data are missing and are to be added to the revised submission.

7.  The reviewer feels that the authors should improve the majority of "results and discussion" section. In spite of the presentation of experimental, analytical and FE work, the paper significantly lacks in-depth explanation and scientific representation of important findings.

8. From Figure 6, please clarify why the peak stress (overall behavior) increases with the decrease in temperature. The mechanism is to be better explained.

Reviewer 3 Report

Review attached

Round 2

Reviewer 2 Report

Most of the comments raised by this reviewer is addressed in the revised version of the paper.

Reviewer 3 Report

Accept in present form